# Common Inflammation-Related Candidate Gene Variants and Acute Kidney Injury in 2647 Critically Ill Finnish Patients

**DOI:** 10.3390/jcm8030342

**Published:** 2019-03-11

**Authors:** Laura M. Vilander, Suvi T. Vaara, Mari A. Kaunisto, Ville Pettilä

**Affiliations:** 1Division of Intensive Care Medicine, Department of Anesthesiology, Intensive Care and Pain Medicine, University of Helsinki and Helsinki University Hospital, 00014 Helsinki, Finland; suvi.vaara@hus.fi (S.T.V.); ville.pettila@hus.fi (V.P.); 2Institute for Molecular Medicine Finland (FIMM), HiLIFE, University of Helsinki, 000014 Helsinki, Finland; mari.kaunisto@helsinki.fi

**Keywords:** acute kidney injury, genetic variation, human genetics

## Abstract

Acute kidney injury (AKI) is a syndrome with high incidence among the critically ill. Because the clinical variables and currently used biomarkers have failed to predict the individual susceptibility to AKI, candidate gene variants for the trait have been studied. Studies about genetic predisposition to AKI have been mainly underpowered and of moderate quality. We report the association study of 27 genetic variants in a cohort of Finnish critically ill patients, focusing on the replication of associations detected with variants in genes related to inflammation, cell survival, or circulation. In this prospective, observational Finnish Acute Kidney Injury (FINNAKI) study, 2647 patients without chronic kidney disease were genotyped. We defined AKI according to Kidney Disease: Improving Global Outcomes (KDIGO) criteria. We compared severe AKI (Stages 2 and 3, *n* = 625) to controls (Stage 0, *n* = 1582). For genotyping we used iPLEX^TM^ Assay (Agena Bioscience). We performed the association analyses with PLINK software, using an additive genetic model in logistic regression. Despite the numerous, although contradictory, studies about association between polymorphisms rs1800629 in *TNFA* and rs1800896 in *IL10* and AKI, we found no association (odds ratios 1.06 (95% CI 0.89–1.28, *p* = 0.51) and 0.92 (95% CI 0.80–1.05, *p* = 0.20), respectively). Adjusting for confounders did not change the results. To conclude, we could not confirm the associations reported in previous studies in a cohort of critically ill patients.

## 1. Introduction

Acute kidney injury (AKI) is a syndrome that often complicates critical illness and is associated with significant mortality and morbidity [1,2]. Thus, efforts to distinguish patients at risk for AKI are justifiable, but despite the advances in the understanding of the pathophysiology of AKI, reliable prediction of developing AKI in different clinical scenarios remains a challenge. 

In our systematic review in 2014 we found that evidence about genetic predisposition to AKI was heterogeneous, the studies were of inadequate size and the findings were generally not replicated [3]. Based on these findings, we analyzed 27 common genetic variants that situate in genes previously associated with AKI in a Finnish sample of critically ill patients. 

## 2. Materials and Methods

### 2.1. Patients

We prospectively recruited patients from 17 Finnish intensive care units (ICUs) in the Finnish acute Kidney Injury (FINNAKI) study. The FINNAKI study took place in the years 2011 and 2012, and the study design has been described previously [1]. We included all patients with an emergency ICU admission of any length and elective surgical patients with an expected duration of ICU stay longer than 24 h. We excluded patients that received maintenance dialysis. The complete exclusion criteria are reported in electronic Appendix A (ESM).

Consent for the study was achieved from the patients or next of kin, at the initiation of the study or deferred. A separate consent for genetic analyses was obtained from all patients or their legal representatives. The Ethics Committee of the Department of Surgery in Helsinki University Hospital gave approval for the study (18/13/03/02/2010). 

### 2.2. Definitions

We defined AKI according to Kidney Disease: Improving Global Outcomes (KDIGO) criteria [4]. We performed analyses using both the severe AKI phenotype (KDIGO Stages 2–3 compared to KDIGO 0) and the all-stage (1–3) AKI phenotype. We classified patients into AKI stages according to daily measurements of plasma creatinine and hourly measurements of urine output. Sepsis was defined according to the American College of Chest Physicians/Society of Critical Care Medicine (ACCP/SCCM) definition [5].

### 2.3. Data Collection

We collected routine data into Finnish Intensive Care Consortium prospective database (Tieto Ltd., Helsinki, Finland). In addition, we completed a standardized case reporting form (CRF) on admission, as well as daily for 5 days and at discharge from ICU. These study-specific data comprised health status previously and present, medications in use, information about some known AKI risk factors, evaluation of organ dysfunctions such as sepsis, and information about treatments administered. 

### 2.4. DNA Samples and Genotyping

Deoxyribonucleic acid (DNA) was extracted from frozen blood samples collected at enrollment. DNA isolation was performed with Chemagic 360 intrument using Chemagic DNA Blood10k Kit, as instructed by the manufacturer (Perkin Elmer, Baesweiler, Germany). For genotyping, we diluted the sample concentration into 10 ng/μL. 

We performed the genotyping in two subsequent assays, in the years 2015 and 2017, at the Genotyping Unit of Institute for Molecular Medicine Finland (FIMM), University of Helsinki. The Agena MassARRAY^®^ system, along with the iPLEXTM Gold Assay (Agena BioscienceTM, San Diego, CA, USA) were used for the genotyping. Here, 20 ng of dried genomic DNA were used for genotyping reactions in 384-well plates using manufacturer’s reagents, and according to their recommendations [6]. For designing primer sequences, MassARRAY Assay Design software (Agena BioscienceTM) was used (see ESM). The MassARRAY Compact System (Agena BioscienceTM) was used for data collection and TyperAnalyzer software (Agena BioscienceTM) for genotype calling. The quality control procedure consisted of checking the success rate, duplicate samples, control wells with water and testing for Hardy–Weinberg Equilibrium (HWE). In addition, all of the genotype calls were manually checked. The genotyping personnel were unaware of the clinical status of the patients.

In the year 2015 assay 49 samples (1.7% of 2968 samples) were rejected because of low success rate in the tested polymorphisms, and in the year 2017 assay the corresponding number was five samples (0.2% of 2968 samples).

For rs699 the success rate reached only 48% due to assay failure in half of the runs; however, the remainder of allele calling was possible with new extension primer and the results are thus reported.

### 2.5. Variant Selection

Variations in or nearby genes related to inflammation, circulation, and cell survival have been suggested in candidate polymorphism studies regarding AKI [7,8,9,10,11,12,13,14,15,16,17,18,19,20,21,22,23,24,25,26,27]. Additionally, the first hypothesis-free studies in AKI genetic predisposition have been published [28,29,30], with some replicated associations [31]. We chose to test polymorphisms in *TNFA* (rs1800629 [8,19,21,22,23,24,25,26,27]), *IL6* (rs1800796 [24,26] and rs1800795 [19,24,26], rs10499563, rs1474347, rs13306435, rs2069842 and rs2069830), *CXCL8* (rs4073 [27]), *IL10* (rs1800896 [19,21,23,25,26]), *NOS3* (rs2070744 [13,24]), *NFKB1A* (rs1050851 [32]), *AGT* (rs699 and rs2493133 [24]), *VEGFA* (rs2010963 and rs3025039 [27]), *EPO* (rs1617640 [14]), *SUFU* (rs10748825 [9]), *HIF1A* (rs11549465 [15]), *PNMT* (rs876493 [17]), *MPO* (rs7208693 [16]), *COMT* (rs4680 [10,11,12]), *HSPB1* (rs2868371 [33]), *SFTPD* (rs2243639 and rs721917 [34]), *HAMP* (rs10421768 [35]) and *BBS9* (rs10262995 [30]) genes (see definitions for abbreviations in ESM).

### 2.6. Statistical Analyses

We used SPSS Statistics version 22 (IBM Corp., Armonk, NY, USA) for analyzing the clinical and demographic variables. The analyses used were the Fisher’s exact test in cross tabulation for categorical variables and the Mann–Whitney U for continuous variables. The data are presented as medians (with interquartile range), or absolute count (with percentage).

We performed the association test for genetic variants and AKI phenotype with logistic regression in the PLINK software [36]. We used the additive genetic model. For haplotype analysis, we checked for haploblocks with Haploview [37]. In addition, for polymorphisms in TNFA and IL10 we performed an epistasis test. The haplotype analysis and epistasis test were performed with the PLINK software [36]. In the primary analysis we compared patients with KDIGO stage 2 or 3 AKI to patients without AKI, as the primary outcome of the study. In the secondary analysis we compared all stage AKI (KDIGO 1, 2 or 3) to no AKI. In the tertiary analysis we included patients with chronic kidney disease and compared patients with KDIGO stage 2 or 3 AKI to patients without AKI in an adjusted analysis. We used similar covariates to our previously published article [31] (liver failure, body mass index (BMI), use of nonsteroidal anti-inflammatory drugs (NSAID) or warfarin as permanent medication, use of contrast dye prior to ICU admission, use of colloids prior to ICU admission, use of albumin prior to ICU admission, minimum platelet count, and simplified acute physiology score (SAPS) II without renal and age points), omitting the infection focus for irrelevant information, and maximum leucocyte count and operative admission to avoid multicollinearity, while including cardiac surgery status as well as sepsis status. The missing data within covariates (altogether 2.4%) were addressed by imputing (see ESM for details).

For all analyses, we considered a *p*-value of 0.002 significant after Bonferroni correction for multiple testing (0.05/25 = 0.002).

### 2.7. Power Calculations

We performed prospective power calculations to determine an appropriate sample size [38]. Assuming an effect size of 1.2 per risk allele (1.4 per homozygote genotype) and a minor allele frequency of 0.2, setting the level of significance to 0.005, there will be 96% (93% for homozygote genotype) power to detect an association in a sample of 1200 cases and 1800 controls.

## 3. Results

We recruited 2968 patients in the FINNAKI genetic study (Figure 1). We excluded 199 patients (6.7% of 2968 patients) with chronic kidney disease from the primary (green dashed line) and secondary (orange dashed line) analysis. In addition, 122 DNA samples failed at isolation. Of the remaining 2647 patients, 221 (8.3% of 2647) patients had stage 2 AKI, 404 (15.3% of 2647) had stage 3 AKI, and 1582 (59.8% of 2647) patients served as controls without AKI. Overall, 228 (10.9% of 2647) patients received renal replacement therapy (RRT). Moreover, 440 (16.6% of 2647) patients had an ambiguous phenotype (KDIGO stage 1); we excluded them from the primary and the tertiary (blue dashed line) analysis. Table 1 presents patient demographics.

In the primary analysis, none of the previously reported associations replicated in our sample (Figure 2). Of note, the A-allele of rs1800629 in TNFA was not associated with AKI (odds ratio, OR 1.06, 95% confidence interval, CI 0.89–1.28, *p* = 0.51). In addition, the G-allele of rs1800896 in IL10 was not associated with AKI (OR 0.92, 95% CI 0.80–1.05, *p* = 0.20) (Table 2). In the epistasis test between A-allele of rs1800629 and G-allele of rs1800896 we detected no evidence of interaction (OR 1.10, *p* = 0.40). Frequencies of genotype combinations of these two variants are presented in Table 3, grouped according to their reported effect on protein production [25].

We tested IL6 for altogether seven variants and found no association with either endpoint. We found no variation in single nucleotide polymorphisms (SNPs) rs2069842 and rs2069830 in IL6; rs10499563 in IL6 was not in HWE (*p* = 0.034). The haplotypes of the two tested haploblocks were not associated with AKI (data shown in ESM). 

The T-allele of rs3025039 in VEGFA had an odds ratio (OR) of 1.20 (95% CI 1.01–1.44, *p* = 0.044) for development of stage 2 or 3 AKI. This finding prevailed in the tertiary analysis (adjusted model, OR 1.21, 95% CI 1.00–1.45, *p* = 0.047; ESM, Appendix A). 

In the tertiary analysis the G-allele of rs10421768 in HAMP had an OR of 0.81 (95% CI 0.69–0.95, *p* = 0.0090) for development of stage 2 or 3 AKI; however, none of the variants had a statistically significant association in this adjusted model with CKD patients included (Appendix A, ESM).

In the secondary analysis with all stage AKI as the endpoint the results did not change (ESM, Appendix A).

The variants we investigated had minor allele frequencies ranging from 0.03 to 0.47 and the retrospectively calculated variant-specific power varied accordingly from 13.0% to 91.2% (ESM, Appendix A).

## 4. Discussion

In this study involving nearly 3000 critically ill adult patients, we were unable to replicate the previously reported associations between selected inflammation-related gene variants and AKI. The 27 tested polymorphisms are within 18 candidate genes, the majority of which relate to inflammation, cell survival, or circulation. Despite the suggestive findings in VEGFA and HAMP, we did not achieve the pre-set statistical significance after correcting for multiple comparisons.

The original studies reporting genetic associations have been generally underpowered, and of moderate quality only [3]. Majority of these studies investigated candidate polymorphisms with unknown biological function. Heterogeneity in reporting has hampered conduction of meta-analyses of reported associations [3,39]. In addition, the few replication attempts have given contradicting results. Moreover, most reports are from cardiac surgery patients, whereas septic and mixed ICU patients have been studied less [3]. In our prospective power calculation we determined that a sample of 1200 cases and 1800 controls would suffice to give a 96% power to detect an association, with realistic assumptions of minor allele frequency and effect size considering complex disease origin. However, the true effects of associations are known to be smaller than the ones reported by first authors [40,41]. In addition, false positive associations are numerous in genetic association studies to identify common variants [41]. This, along with population diversity is a possible explanation as to the failure in replication attempt [42]. Because of multiple variants tested, we used a more stringent level of significance. By using Bonferroni method we determined the acceptable level of type 1 error rate to 0.002.

The first results of hypothesis-free study designs in genetic predisposition to AKI have been published in septic [28] and cardiac surgery-associated AKI [29,30]. In their genome wide association study (GWAS), Stafford-Smith and coworkers [30] reported an association of polymorphism rs10262995 in BBS9 with cardiac surgery-associated AKI. However, in our cohort this association was not found. 

One of the most studied polymorphisms is the rs1800629 in TNFA: the low producing genotype (GG) has been associated with more frequent and more severe AKI [7,8]. In addition, low producing genotype AA of variant rs1800896 in IL10 has been associated with AKI [7], along with combined genotype of rs1800629 GG + rs1800896 AA [19]. However, contradicting findings have been presented [22,23,24,25,26]. Additionally, TNFA and IL10 variations have been associated with sepsis development [43,44,45]. Of note, we were unable to detect any significant association between these polymorphisms or their combination in the epistasis test. 

Vascular endothelial growth factor (VEGF) is a protein with shown effects in angiogenesis, cell survival and differentiation, as well as vascular permeability [46,47]. In our study, the T-allele in rs3025039 in VEGFA resulted in an OR of 1.20 for AKI (95% CI 1.01–1.44, *p* = 0.044); however, previously the C-allele has been reported to increase AKI risk [27]. In carriers of T-allele, the plasma VEGF levels are lower [48]. The T-allele has been suggested to relate to susceptibility to ARDS and mortality in ARDS [49,50]. Additionally, variation in VEGFA has been associated with diabetic nephropathy [51].

As the IL-6 cytokine has been associated with AKI development [52,53], we aimed to investigate the IL6 gene variation more broadly. We genotyped seven SNPs, five in addition to the two replication variants. However, two of these SNPs did not have any variation in our sample. The remaining five did not associate with AKI, even when studied as a haplotype. In sepsis, the SNP rs1800795 is not associated with susceptibility or mortality [54]. Consistently, rs1800795 does not correlate with the risk of end-stage renal disease [55]. Nevertheless, in patients with CKD, the SNP rs1800796 is suggested to predispose to sepsis and mortality [56]. However, due to multiple differing etiologies the predisposing genetic variants are generally unique to CKD [57,58,59].

The rs4680 in catechol-O-methyltransferase gene (COMT) causes an amino acid transition (Val158Met), which leads to lower (L) in comparison to higher (H) enzyme activity [60]. The COMT enzyme degrades catecholamines [61], and thus has been thought to contribute in vasodilatory shock and AKI [11]. The LL genotype is associated in cardiac surgery associated AKI (CSA-AKI) in two studies of modest size [11,12], yet in a larger study this association was ruled out [10]. Furthermore, endothelial NO synthase gene (NOS3) variant rs2070744 was studied in association to CSA-AKI with conflicting results [13,24]. The rs2070744 has been investigated in association with diabetic nephropathy, however, results have contradicted [62]. We did not find any association between these variants and AKI. 

To our knowledge, this is the first replication of polymorphisms in *CXCL8* [27], *NFKB1A* [32], *AGT* [24], *EPO* [14], *SUFU* [9], *HIF1A* [15], *PNMT* [17], *MPO* [16], and *SFTPD* [34] in association to AKI phenotype. None of the investigated variants rendered verification for the initial hypotheses.

In addition to the candidate polymorphism replication, we tested two SNPs due to interesting biological hypotheses. Li and colleagues [33] presented in their in vitro septic AKI model that heat shock protein 27 (Hsp27) overexpression caused the renal epithelial cells to outlive. The C-allele of a functional SNP rs2868371 in the HSPB1 gene associates with decreased expression of Hsp27 [63,64]. However, in our sample this SNP was not associated with AKI in any of the analyses. Another intriguing suggestion regarding the pathophysiology of septic AKI was presented by Schaalan and colleagues [35]: the hepcidin levels were elevated in patients with septic AKI. Hepcidin is encoded by HAMP gene, and the promoter SNP rs10421768 is suggested to affect the gene expression [65,66]. We found G-allele to be protective (OR 0.81, 95% CI 0.69–0.95, *p* = 0.0090) in the adjusted model; however, this was not a replication of an association to a human AKI model, but a pilot study on this association.

We acknowledge that our study has some limitations. First, we were unable to extract the DNA of 122 (4.1%) patients. However, this random selection is unlikely to cause bias in our remaining data.

Second, our cohort of critically ill patients consists of patients with multiple possible etiologies for AKI, rather than tightly defined phenotypes, such as cardiac surgery or sepsis. However, we did adjust for these confounders in our tertiary analysis.

Third, the actual sample size was somewhat smaller than the estimated sample size we prospectively estimated to be needed for an adequate power. However, even with samples of 2207 patients in primary analysis, 2647 patients in the secondary analysis, and 2358 patients in the tertiary analysis, the retrospectively calculated power of the study, holding to the presumptions about allele frequency and effect size, remained adequate (80.6%, 93.6% and 85.5%, respectively). The minor allele frequency was, however, lower for some of the studied SNPs, affecting the power of these specific analyses. Most SNPs had frequencies exceeding the 0.2 we anticipated. A larger sample of patients with a sub-phenotype such as septic AKI is an interesting challenge for the future.

## 5. Conclusions

In conclusion, we were unable to replicate previous associations between genetic variants and AKI in critically ill patients. Even if short of significance, an interesting previously unpublished variant in the HAMP gene offers possible insight into mechanism of AKI, although future studies are needed to confirm this finding. In the future, the efforts to decipher “the AKI gene” should be targeted on more carefully assigned AKI sub-phenotypes.

## Figures and Tables

**Figure 1 jcm-08-00342-f001:**
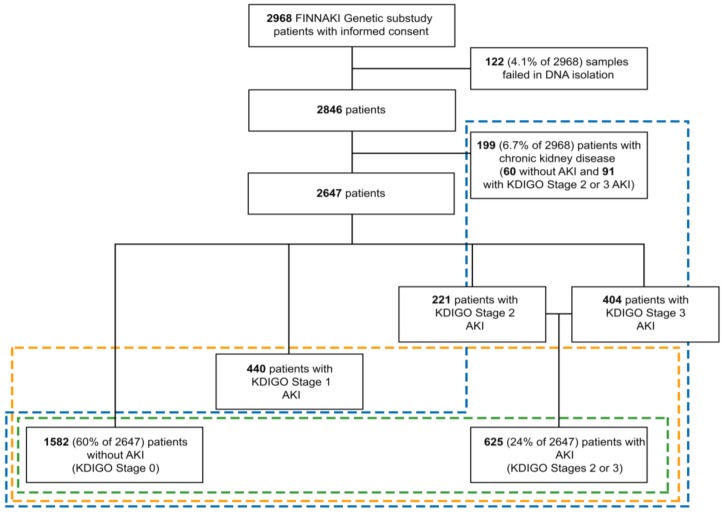
Study flowchart. Abbreviations: FINNAKI; Finnish Acute Kidney Injury; DNA, deoxyribonucleic acid; AKI, acute kidney injury; KDIGO, Kidney Disease: Improving Global Outcomes.

**Figure 2 jcm-08-00342-f002:**
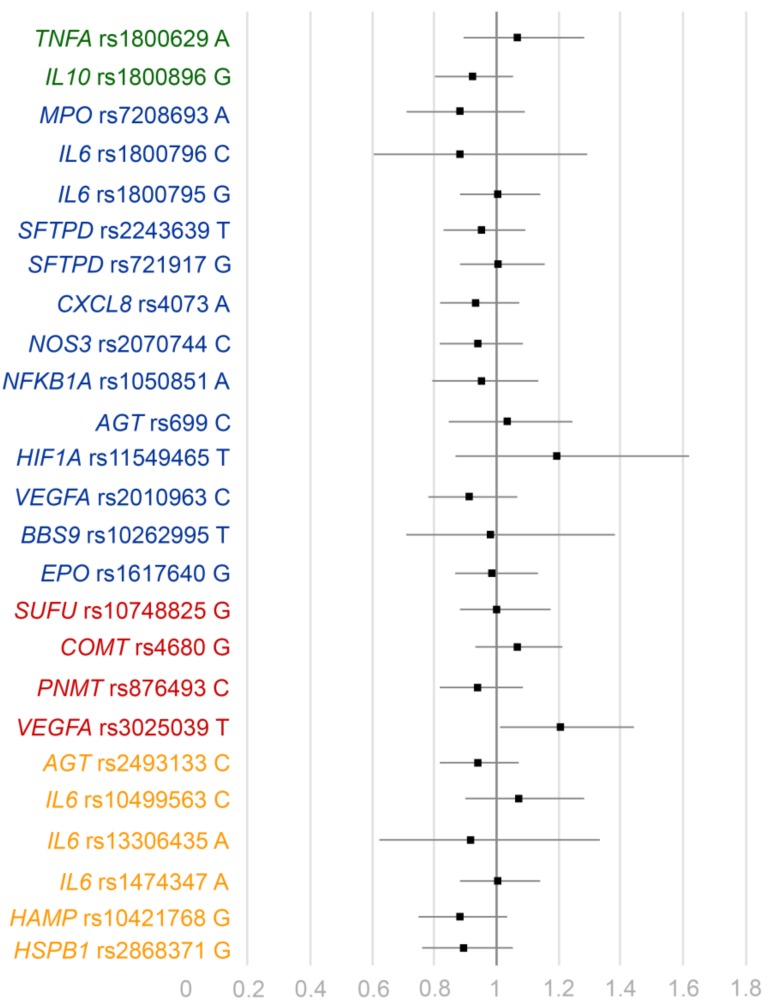
Odds ratios (OR) and confidence intervals (95% CI) for the minor allele for all the studied polymorphisms. For variants in green, there are several previous studies and both alleles have been reported to associate with AKI, variants in blue have been reported with same risk allele, variants in red have been reported with opposite risk allele, and variants in orange have not been previously reported in association to AKI.

**Table 1 jcm-08-00342-t001:** Demographics of altogether 2647 patients in the FINNAKI genetic substudy after excluding patients with maintenance dialysis. Data are presented according to presence of severe AKI (KDIGO stage 2 or 3, *n* = 625), presence of all stage AKI (KDIGO stage 1, 2, 3, *n* = 1065), or absence of AKI (KDIGO stage 0, *n* = 1582).

Characteristics	Data Available	AKI		No AKI	*p* *
		KDIGO stage 2 or 3	KDIGO stage 1, 2, 3		
Age (years)	2647	65 (54–74)	65 (54–75)	62 (48–72)	<0.001
Gender (male)	2647	409 (65.4)	700 (65.7)	980 (61.9)	0.130
BMI (kg/m^2^)	2627	27.5 (24.5–31.3)	27.5 (24.2–31.2)	25.7 (23.1–28.7)	<0.001
Co-morbidities					
Arterial hypertension	2633	333 (53.3)	561 (52.9)	641 (40.8)	<0.001
Diabetes	2643	169 (27.0)	263 (24.7)	280 (17.7)	<0.001
Arteriosclerosis	2623	94 (15.1)	159 (15.0)	160 (10.2)	0.002
Chronic obstructive pulmonary disease	2630	43 (6.9)	81 (7.7)	136 (8.6)	0.195
Chronic liver disease	2617	46 (7.4)	59 (5.6)	51 (3.3)	<0.001
Systolic heart failure	2628	79 (12.7)	129 (12.2)	139 (8.8)	0.009
Baseline plasma creatinine (µmol/L)	2643	81.0 (68.9–94.0)	81.0 (69.0–94.0)	79.0 (68.0–94.0)	0.210
Pre ICU daily medication					
ACE inhibitor or ARB	2585	263 (42.8)	428 (41.1)	475 (30.8)	<0.001
NSAID	2538	73 (12.1)	112 (10.9)	118 (7.8)	0.002
Diuretic	2596	185 (29.8)	324 (30.8)	323 (20.9)	<0.001
Metformin	2606	109 (17.6)	163 (15.5)	164 (10.6)	<0.001
Statin	2603	196 (31.6)	320 (30.5)	397 (25.6)	0.005
Corticosteroids	2614	56 (9.0)	94 (8.9)	105 (6.7)	0.070
Warfarin	2608	107 (17.2)	166 (15.8)	179 (11.5)	0.001
Treatments administered 48 h before admission					
Contrast medium	2632	120 (19.3)	223 (21.1)	417 (26.5)	<0.001
ACE inhibitor or ARB	2601	167 (27.3)	287 (27.5)	329 (21.1)	0.002
Diuretics	2570	217 (35.8)	353 (34.3)	360 (23.4)	<0.001
Colloids (gelatin or starch)	2479	229 (38.3)	395 (39.0)	394 (26.9)	<0.001
Albumin	2584	14 (2.3)	18 (1.7)	14 (0.9)	0.018
Type of admission					
Operative	2646	180 (28.8)	343 (32.2)	557 (35.2)	0.004
Cardiac surgery	2647	35 (5.6)	80 (7.5)	147 (9.3)	0.004
Emergency	2621	575 (92.6)	962 (91.1)	1386 (88.6)	0.005
SAPS II score 24 h without renal or age components	2614	24.0 (16.0–34.0)	24.0 (16.0–32.0)	20.0 (13.0–29.3)	<0.001
Mechanical ventilation	2647	432 (69.1)	776 (72.9)	1031 (65.2)	0.080
Sepsis	2647	309 (49.4)	500 (46.9)	362 (22.9)	<0.001
White blood cell count at admission, max (10^9^/L)	2186	12.0 (8.3–17.4)	11.7 (8.2–16.8)	10.9 (7.8–15.3)	<0.001
Platelet count at admission, min (10^9^/L)	2419	190.0 (116.5–263.5)	194.0 (127.0–265.0)	205.0 (153.0–268.0)	<0.001

* Comparison of No AKI to KDIGO stages 2 or 3 AKI. The *p*-values are calculated with Fisher’s exact test for categorical variables and with Mann–Whitney U test for continuous variables. Data presented as medians and interquartile ranges for continuous variables, and absolute counts and percentages for categorical variables. Abbreviations: FINNAKI; Finnish Acute Kidney Injury; AKI, acute kidney injury; KDIGO, Kidney Disease: Improving Global Outcomes; BMI, body mass index; ICU, intensive care unit; ACE, angiotensin-converting enzyme; ARB, angiotensin II receptor blocker; NSAID, nonsteroidal anti-inflammatory drug; SAPS II, simplified acute physiology score II.

**Table 2 jcm-08-00342-t002:** Association of genetic variants with acute kidney injury (AKI) KDIGO stages 2 and 3 compared to stage 0. Odds ratios (OR) and confidence intervals (95% CI) are reported for each copy of minor allele.

Gene	SNP	Patients	Minor Allele	MAF (Cases/Controls)	Additive Logistic OR	95% CI	*p*
*TNFA*	rs1800629	2174	A	0.15/0.14	1.06	0.89–1.28	0.51
*IL10*	rs1800896	2173	G	0.44/0.46	0.92	0.80–1.05	0.20
*IL6*	rs10499563	2192	C	0.15/0.14	1.07	0.90–1.28	0.45
	rs1800796	2197	C	0.03/0.03	0.88	0.60–1.29	0.51
	rs1800795	2189	G	0.47/0.47	1.00	0.88–1.14	0.97
	rs1474347	2187	A	0.47/0.47	1.00	0.88–1.14	1.00
	rs13306435	2199	A	0.03/0.03	0.91	0.62–1.33	0.62
*CXCL8*	rs4073	2193	A	0.42/0.42	0.93	0.82–1.07	0.31
*NOS3*	rs2070744	2174	C	0.34/0.36	0.94	0.82–1.08	0.37
*NFKB1A*	rs1050851	2196	A	0.16/0.17	0.95	0.79–1.13	0.54
*AGT*	rs699	1047	C	0.43/0.42	1.03	0.85–1.24	0.78
	rs2493133	2196	C	0.41/0.42	0.94	0.82–1.07	0.36
*VEGFA*	rs2010963	2170	C	0.22/0.24	0.91	0.78–1.06	0.22
	rs3025039	2195	T	0.16/0.14	1.20	1.01–1.44	0.044
*EPO*	rs1617640	2173	G	0.44/0.45	0.99	0.87–1.13	0.91
*SUFU*	rs10748825	2174	G	0.37/0.37	1.02	0.88–1.17	0.83
*HIF1A*	rs11549465	2173	T	0.05/0.04	1.19	0.87–1.62	0.28
*PNMT*	rs876493	2173	C	0.36/0.38	0.94	0.82–1.08	0.37
*MPO*	rs7208693	2174	A	0.10/0.12	0.88	0.71–1.09	0.23
*COMT*	rs4680	2173	G	0.47/0.45	1.06	0.93–1.21	0.39
*HSPB1*	rs2868371	2194	G	0.19/0.21	0.89	0.76–1.05	0.18
*SFTPD*	rs2243639	2199	T	0.39/0.40	0.95	0.83–1.09	0.47
	rs721917	2193	G	0.39/0.39	1.00	0.88–1.15	0.97
*HAMP*	rs10421768	2199	G	0.23/0.25	0.88	0.75–1.03	0.11
*BBS9*	rs10262995	2200	T	0.04/0.04	0.98	0.71–1.38	0.93

Abbreviations: AKI, acute kidney injury; KDIGO, Kidney Disease: Improving Global Outcomes; SNP, single nucleotide polymorphism; MAF, minor allele frequency; OR, odds ratio; CI, confidence interval; *TNFA*, tumor necrosis factor alpha; *IL10*, interleukin 10; *IL6*, interleukin 6; *CXCL8*, interleukin 8; *NOS3*, nitric oxide synthase 3; *NFKB1A*, nuclear factor of kappa light polypeptide gene enhancer in B-cells inhibitor, alpha; *AGT*, angiotensinogen; *VEGFA*, vascular endothelial growth factor; *EPO*, erythropoietin; *SUFU*, suppressor of fused homolog; *HIF1A*, hypoxia-inducible factor 1-alpha; *PNMT*, phenylethanolamine N-methyltransferase; *MPO*, myeloperoxidase; *COMT*, catechol-O-methyltransferase; *HSPB1*, heat shock protein family B (small) member 1; *SFTPD*, surfactant protein D; *HAMP*, hepcidin antimicrobial peptide; *BBS9*, Bardet–Biedl syndrome 9.

**Table 3 jcm-08-00342-t003:** Number of patients (percentage) with *TNFA* rs1800629 and *IL10* rs1800896 genotype combinations. Genotypes are grouped according to the reported effect in protein production. Acute kidney injury (AKI) KDIGO stages 2 and 3 (cases) compared to stage 0 (controls).

Genotype Combination	AKI (*n* = 615)	No AKI (*n* = 1558)
*TNFA* GG + *IL10* AA	138 (22%)	340 (22%)
*TNFA* GG + *IL10* GA + GG	311 (51%)	814 (52%)
*TNFA* GA + AA + *IL10* AA	52 (8%)	116 (8%)
*TNFA* GA + AA + *IL10* GA + GG	114 (19%)	288 (18%)

Abbreviations: TNFA, tumor necrosis factor alpha; IL10, interleukin 10; AKI, acute kidney injury; KDIGO, Kidney Disease: Improving Global Outcomes; OR, odds ratio. TNFA GG: TNF-α low producer; IL10 AA: IL-10 low producer; IL10 GA + GG: IL-10 intermediate + high producer; TNFA GA + AA: TNF-α high producer.

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
