# Peer review of "Common Inflammation-Related Candidate Gene Variants and Acute Kidney Injury in 2647 Critically Ill Finnish Patients"

_jcm, 2019, doi:10.3390/jcm8030342_

Round 1
Reviewer 1 Report
The authors test 27 polymorphisms for their association with AKI taking advantage of stored samples from the FINNAKI study from 2013. Variants selection was guided by a systematic review published in 2014 and the presented analysis deals with shortcomings of previous reports. Samples from 2647 patients were included and completeness was higher than 95% for all except one SNP. None of the variants associated significantly with AKI in any of the planned analyses. The study by Vilander is the most extensive so far in a cohort of mixed etiologies. It is a well-executed investigation.
I have some minor criticisms regarding the statistical analysis section where the authors mention tertiary analysis that 'included patients with chronic kidney disease and compared patients with KDIGO stage 2 or 3 AKI to patients without AKI in an adjusted analysis.' There is no evidence of such analysis in the paper or supplemental data, and samples from patients with CKD appear excluded in Figure 1. It is also not clear which analysis was adjusted for the listed covariates and why age wasn't included, as one of them. Finally, Table 1 should include demographics of AKI stage 1-3.
Author Response
Authors’ response to reviewers’ comments -1
Manuscript ID: jcm-448415
Authors’ response to Reviewer 1 comments:
We thank reviewer1 for their valuable comments to improve our manuscript: Common inflammation-related candidate gene variants and acute kidney injury in 2647 critically ill Finnish patients.
We would like to respond point-by-point to the reviewer report as follows:
Reviewer1 comment 1: There is no evidence of such tertiary analysis that included patients with chronic kidney disease and compared patients with KDIGO stage 2 or 3 AKI to patients without AKI in an adjusted analysis in the paper or supplemental data.
Authors’ response to reviewer1 comment 1: The results of the analysis are presented in the Supplementary file titled: “Results for analyses 2 and 3, Table 4. Adjusted analysis; association of genetic variants with acute kidney injury (AKI) KDIGO Stages 2 and 3 compared to Stage 0. Patients with chronic kidney disease included. Odds ratios (OR) and confidence intervals (95% CI) are reported for each copy of minor allele.” Additionally, we attempted to clarify the findings of the tertiary analysis in the results section (page 9, rows 338 to 340) of the main article.
Reviewer1 comment 2: Samples from patients with CKD appear excluded in Figure 1.
Authors’ response to reviewer1 comment 2: We have redrawn Figure 1 to better illustrate the handling of patients with CKD.
Reviewer1 comment 3: It is also not clear which analysis was adjusted for the listed covariates and why age wasn't included, as one of them.
Authors’ response to reviewer1 comment 3: On page 3 of 14, on rows 112 through 118 we describe the tertiary analysis and the covariates. Age was not included because we closely followed the previous multivariant analysis in the same cohort. The covariates that were found to remain associated in previous multivariate analysis were included (Vilander LM, Kaunisto MA, Vaara ST, Pettilä V; FINNAKI Study Group. Genetic variants in SERPINA4 and SERPINA5, but not BCL2 and SIK3 are associated with acute kidney injury in critically ill patients with septic shock. Critical Care 2017, 21, 47/Methods page 3 of 11).
Reviewer1 comment 4: Table 1 should include demographics of AKI stage 1-3.
Authors’ response to reviewer1 comment 4: We added a column with covariates for patients with AKI.
On behalf of the authors,
Laura M. Vilander
Ville Pettilä
Reviewer 2 Report
Review comments on Vilander et al. “Common inflammation-related candidate gene variants and acute kidney injury in 2647 critically ill Finnish patients”.
Vilander et al. present a genetic association study in a cohort of 2647 critically ill patients including 27 gene variants previously reported to be associated with acute kidney injury (AKI). The authors genotyped 27 SNPs in the cohort of the FINNAKI study (published in 2013 and investigating the incidence and 90-day mortality of AKI in 2901 patients from 17 Finnish intensive care units) and compared severe AKI (stage 2 and 3) to no AKI (stage 0). The authors could not confirm the genetic associations reported in previous studies.
The topic of this study is of experimental and potential clinical interest, the manuscript is well structured, but important methodological weaknesses substantially reduce the impact of the manuscript.
MAJOR POINTS:
- The main issue is related to the power calculation. The estimated effect size of 1.2 per risk allele seems quite high for a clinical study focused on a multifactorial process, such as AKI, and including a heterogeneous study population. Even if we would accept this estimation, the power calculation indicated that 3000 patients needed to be enrolled. Unfortunately, the number of patients eventually included in the primary outcome analysis was 2207. Thus, the study is underpowered, and the correct interpretation of negative results is extremely challenging. Importantly, the data might be misinterpreted by the readers, since it is not possible to exclude that the negative results presented in the study are only the consequence of an insufficient number of patients.
- Starting from the database of the FINNAKI study it should be possible for the authors to perform additional analyses to investigate genetic associations with the outcome after AKI. This might provide important information about the response to AKI in humans.
MINOR POINTS
- The definition of CKD should be revised: the authors “excluded patients that received maintenance dialysis for chronic kidney disease (CKD)”. This does not correspond to CKD but to end stage renal disease (ESRD). Moreover, the authors should present baseline renal function among the characteristics of the study population and – if our second major point will be considered – stratify the long-term outcome analysis according to baseline renal function.
- Figure 2 should be re-designed to improve readability. We strongly suggest dividing the figures into two parts by separating the minor alleles previously associated with a positive and a negative effect on AKI, respectively. This will help to evaluate if at least the trend was the same as in previous studies.
Author Response
Authors’ response to reviewers’ comments -2
Manuscript ID: jcm-448415
Authors’ response to Reviewer 2 comments:
We thank reviewer2 for valuable comments to improve our manuscript: Common inflammation-related candidate gene variants and acute kidney injury in 2647 critically ill Finnish patients.
We would like to respond point-by-point to the reviewer report as follows:
Reviewer2 major comment 1:
The main issue is related to the power calculation. The estimated effect size of 1.2 per risk allele seems quite high for a clinical study focused on a multifactorial process, such as AKI, and including a heterogeneous study population. Even if we would accept this estimation, the power calculation indicated that 3000 patients needed to be enrolled. Unfortunately, the number of patients eventually included in the primary outcome analysis was 2207. Thus, the study is underpowered, and the correct interpretation of negative results is extremely challenging. Importantly, the data might be misinterpreted by the readers, since it is not possible to exclude that the negative results presented in the study are only the consequence of an insufficient number of patients.
Authors’ response to reviewer2 major comment 1:
Reviewer2 raises the question of adequate power and finds the study underpowered. The authors want to emphasize that in the statistical methods (page 3, rows 124 to 127) we described the prospective power calculations. We have now also performed retrospective power calculations and report these results in the manuscript. It should also be noted that the sentence about power calculations in Discussion (page 9, rows 357 to 358) was actually incorrect and we apologize for that. Using the prospective sample size estimation, 3000 patients, the power exceeds 95%. We have now corrected this information.
The retrospective power calculations we performed show that for analysis 1, where there were 625 cases and 1582 controls, using the same assumptions as for the prospective power calculations (effect size of 1.2 per risk allele and minor allele frequency of 0.2), the power would be 80.6%. We consider this an adequate power (normally 80% considered adequate), although less than prospectively anticipated.
Furthermore, it should be noted that the median number of patients in the AKI candidate gene studies published until 2017 was only 262 (range 61 to 1741) as shown in our systematic review (Vilander LM, Kaunisto MA, Pettilä V. Genetic predisposition to acute kidney injury – a systematic review. BMC Nephrology 2015, 16, 1–10). Thus, our study exceeds all these published original studies in sample size. Additionally, reviewer 1 assessed our study as “the most extensive so far in a cohort of mixed etiologies”.
Based on reviewer comment, we added in retrospectively calculated power for each variant (Supplementary material, Table 6) and we mention in the main text the range for these power calculations (page 9, rows 343 to 345). We have edited the paragraph in the Discussion about study strengths (page 9, rows 357 to 359). We added a paragraph in the Discussion where we discuss the sample size and the power as one of the limitations of our study (page 11, rows 429 to 430). We discuss the retrospectively calculated power, along with remark about the larger samples in sub-phenotypes. Additionally, we provide the CIs for ORs for each risk allele (Figure 2) to give the reader a clear view for interpretation of possibility of type 2 error.
Reviewer2 major comment 2:
Starting from the database of the FINNAKI study it should be possible for the authors to perform additional analyses to investigate genetic associations with the outcome after AKI. This might provide important information about the response to AKI in humans.
Authors’ response to reviewer2 major comment 2: We thank the reviewer for this interesting idea. Regrettably no such data are available that would combine the long-term outcome of patients in FINNAKI genetic sub-study. Our predefined outcome in this study was development of AKI and, thus, we followed the predefined analysis plan.
Reviewer2 minor comment 1:
The definition of CKD should be revised: the authors “excluded patients that received maintenance dialysis for chronic kidney disease (CKD)”. This does not correspond to CKD but to end stage renal disease (ESRD). Moreover, the authors should present baseline renal function among the characteristics of the study population and – if our second major point will be considered – stratify the long-term outcome analysis according to baseline renal function.
Authors’ response to reviewer2 minor comment 1:
We have revised the incorrect sentence into “We excluded patients that received maintenance dialysis”. In table 1 we present the baseline plasma creatinine for patients with and without AKI.
Reviewer2 minor comment 2:
Figure 2 should be re-designed to improve readability. We strongly suggest dividing the figures into two parts by separating the minor alleles previously associated with a positive and a negative effect on AKI, respectively. This will help to evaluate if at least the trend was the same as in previous studies.
Authors’ response to reviewer2 minor comment 2:
We thank for the suggestion and have redrawn Figure 2 accordingly. We use color coding and grouping of the alleles and agree that it is now more informative and easily readable.
On behalf of the authors,
Laura M. Vilander
Ville Pettilä
Round 2
Reviewer 2 Report
None.